# Robust Cross-lingual Embeddings from Parallel Sentences

## Abstract

Recent advances in cross-lingual word embeddings have primarily relied on mapping-based methods, which project pre-trained word embeddings from different languages into a shared space through a linear transformation. However, these approaches assume word embedding spaces are isomorphic between different languages, which has been shown not to hold in practice (Søgaard et al., 2018), and fundamentally limits their performance. This motivates investigating joint learning methods which can overcome this impediment, by simultaneously learning embeddings across languages via a cross-lingual term in the training objective. Given the abundance of parallel data available (Tiedemann, 2012), we propose a bilingual extension of the CBOW method which leverages sentence-aligned corpora to obtain robust cross-lingual word and sentence representations. Our approach significantly improves cross-lingual sentence retrieval performance over all other approaches, as well as convincingly outscores mapping methods while maintaining parity with jointly trained methods on word-translation. It also achieves parity with a deep RNN method on a zero-shot cross-lingual document classification task, requiring far fewer computational resources for training and inference. As an additional advantage, our bilingual method also improves the quality of monolingual word vectors despite training on much smaller datasets. We make our code and models publicly available.

## 1 Introduction

Cross-lingual representations—such as embeddings of words and phrases into a single comparable feature space—have become a key technique in multilingual natural language processing. They offer strong promise towards the goal of a joint understanding of concepts across languages, as well as for enabling the transfer of knowledge and machine learning models between different languages. Therefore, cross-lingual embeddings can serve a variety of downstream tasks such as bilingual lexicon induction, cross-lingual information retrieval, machine translation and many applications of zero-shot transfer learning, which is particularly impactful from resource-rich to low-resource languages.

Existing methods can be broadly classified into two groups (Ruder et al., 2017): *mapping methods* leverage existing monolingual embeddings which are treated as independent, and apply a post-process step to map the embeddings of each language into a shared space, through a linear transformation (Mikolov et al., 2013b; Conneau et al., 2017; Joulin et al., 2018). On the other hand, *joint methods* learn representations concurrently for multiple languages, by combining monolingual and cross-lingual training tasks (Luong et al., 2015; Coulmance et al., 2015; Gouws et al., 2015; Vulic & Moens, 2015; Chandar et al., 2014; Hermann & Blunsom, 2013).

While recent work on word embeddings has focused almost exclusively on mapping methods, which require little to no cross-lingual supervision, (Søgaard et al., 2018) establish that their performance is hindered by linguistic and domain divergences in general, and for distant language pairs in particular. Principally, their analysis shows that cross-lingual hubness, where a few words (hubs) in the source language are nearest cross-lingual neighbours of many words in the target language, and structural non-isometry between embeddings do impose a fundamental barrier to the performance of linear mapping methods.

(Ormazabal et al., 2019) propose using joint learning as a means of mitigating these issues. Given parallel data, such as sentences, a joint model learns to predict either the word or context in both

source and target languages. As we will demonstrate with results from our algorithm, joint methods yield compatible embeddings which are closer to isomorphic, less sensitive to hubness, and perform better on cross-lingual benchmarks.

**Contributions.** We propose the BI-SENT2VEC algorithm, which extends the SENT2VEC algorithm (Pagliardini et al., 2018; Gupta et al., 2019) to the cross-lingual setting. We also revisit TRANS-GRAM Coulmance et al. (2015), another joint learning method, to assess the effectiveness of joint learning over mapping-based methods. Our contributions are

- On cross-lingual sentence-retrieval and monolingual word representation quality evaluations, BI-SENT2VEC significantly outperforms competing methods, both jointly trained as well as mapping-based ones while preserving state-of-the-art performance on cross-lingual word retrieval tasks. For dis-similar language pairs, BI-SENT2VEC outperform their competitors by an even larger margin on all the tasks hinting towards the robustness of our method.

- BI-SENT2VEC performs on par with a multilingual RNN based sentence encoder, LASER (Artetxe & Schwenk, 2018), on MLDoc (Schwenk & Li, 2018), a zero-shot cross-lingual transfer task on documents in multiple languages. Compared to LASER, our method improves computational efficiency by an order of magnitude for both training and inference, making it suitable for resource or latency-constrained on-device cross-lingual NLP applications.

- We verify that joint learning methods consistently dominate state-of-the-art mapping methods on standard benchmarks, i.e., cross-lingual word and sentence retrieval.

- Training on parallel data additionally enriches monolingual representation quality, evident by the superior performance of BI-SENT2VEC over FASTTEXT embeddings trained on a $100\times$ larger corpus.

We make our models and code publicly available.

## 2 RELATED WORK

The literature on cross-lingual representation learning is extensive. Most recent advances in the field pursue unsupervised (Artetxe et al., 2017; Conneau et al., 2017; Chen & Cardie, 2018; Hoshen & Wolf, 2018; Grave et al., 2018b) or supervised (Joulin et al., 2018; Conneau et al., 2017) mapping or alignment-based algorithms. All these methods use existing monolingual word embeddings, followed by a cross-lingual alignment procedure as a post-processing step— that is to learn a simple (typically linear) mapping from the source language embedding space to the target language embedding space.

Supervised learning of a linear map from a source embedding space to another target embedding space (Mikolov et al., 2013b) based on a bilingual dictionary was one of the first approaches towards cross-lingual word embeddings. Additionally enforcing orthogonality constraints on the linear map results in rotations, and can be formulated as an orthogonal Procrustes problem (Smith et al., 2017). However, the authors found the translated embeddings to suffer from hubness, which they mitigate by introducing the *inverted softmax* as a corrective search metric at inference time. (Artetxe et al., 2017) align embedding spaces starting from a parallel seed lexicon such as digits and iteratively build a larger bilingual dictionary during training.

In their seminal work, (Conneau et al., 2017) propose an adversarial training method to learn a linear orthogonal map, avoiding bilingual supervision altogether. They further refine the learnt mapping by applying the Procrustes procedure iteratively with a synthetic dictionary generated through adversarial training. They also introduce the 'Cross-Domain Similarity Local Scaling' (CSLS) retrieval criterion for translating between spaces, which further improves on the word translation accuracy over nearest-neighbour and inverted softmax metrics. They refer to their work as *Multilingual Unsupervised and Supervised Embeddings (MUSE)*. In this paper, we will use MUSE to denote the unsupervised embeddings introduced by them, and "*Procrustes + refine*" to denote the supervised embeddings obtained by them. (Chen & Cardie, 2018) similarly use "multilingual adversarial training" followed by "pseudo-supervised refinement" to obtain *unsupervised multilingual word embeddings (UMWE)*, as opposed to bilingual word embeddings by (Conneau et al., 2017). Hoshen & Wolf (2018) describe an unsupervised approach where they align the second moment of the two

word embedding distributions followed by a further refinement. Building on the success of CSLS in reducing retrieval sensitivity to hubness, (Joulin et al., 2018) directly optimize a convex relaxation of the CSLS function (*RCSLS*) to align existing mono-lingual embeddings using a bilingual dictionary.

While none of the methods described above require parallel corpora, all assume structural isomorphism between existing embeddings for each language (Mikolov et al., 2013b), i.e. there exists a simple (typically linear) mapping function which aligns all existing embeddings. However, this is not always a realistic assumption (Søgaard et al., 2018)—even in small toy-examples it is clear that many geometric configurations of points can not be linearly mapped to their targets.

Joint learning algorithms such as TRANSGRAM (Coulmance et al., 2015) and Cr5 (Josifoski et al., 2019) , circumvent this restriction by simultaneously learning embeddings as well as their alignment. TRANSGRAM, for example, extends the Skipgram (Mikolov et al., 2013a) method to jointly train bilingual embeddings in the same space, on a corpus composed of parallel sentences. In addition to the monolingual Skipgram loss for both languages, they introduce a similar cross-lingual loss where a word from a sentence in one language is trained to predict the word-contents of the sentence in the other. Cr5, on the other hand, uses document-aligned corpora to achieve state-of-the-art results for cross-lingual document retrieval while staying competitive at cross-lingual sentence and word retrieval. TRANSGRAM embeddings have been absent from discussion in most of the recent work. However, the growing abundance of sentence-aligned parallel data (Tiedemann, 2012) merits a reappraisal of their performance.

(Ormazabal et al., 2019) use BIVEC (Luong et al., 2015), another bilingual extension of Skipgram, which uses a bilingual dictionary in addition to parallel sentences to obtain word-alignments and compare it with the unsupervised version of VECMAP (Artetxe et al., 2018b), another mapping-based method. Our experiments show this extra level of supervision in the case of BIVEC is redundant in obtaining state-of-the-art performance.

## 3 MODEL

**Proposed Model.** Our BI-SENT2VEC model is a cross-lingual extension of SENT2VEC proposed by (Pagliardini et al., 2018), which in turn is an extension of the *C-BOW* embedding method (Mikolov et al., 2013a). SENT2VEC is trained on sentence contexts, with the word and higher-order word n-gram embeddings specifically optimized toward obtaining robust sentence embeddings using additive composition. Formally, SENT2VEC obtains representation $\boldsymbol{v}_s$ of a sentence $S$ by averaging the word-ngram embeddings (including unigrams) as $\boldsymbol{v}_s := \frac{1}{R(S)} \sum_{w \in R(S)} \boldsymbol{v}_w$ where $R(S)$ is the set of word n-grams in the sentence $S$.

The SENT2VEC training objective aims to predict a masked word token $w_t$ in the sentence $S$ using the rest of the sentence representation $\boldsymbol{v}_{S \setminus \{w_t\}}$. To formulate the training objective, we use logistic loss $\ell : x \mapsto \log\left(1 + e^{-x}\right)$ in conjunction with negative sampling. More precisely, for a raw text corpus $\boldsymbol{C}$, the monolingual training objective for SENT2VEC is given by

$$\min_{\boldsymbol{U}, \boldsymbol{V}} \sum_{S \in \boldsymbol{C}} \sum_{w_t \in S} \left( \ell\big(\boldsymbol{u}_{w_t}^\top \boldsymbol{v}_{S \setminus \{w_t\}}\big) + \sum_{w' \in N_{w_t}} \ell\big(-\boldsymbol{u}_{w'}^\top \boldsymbol{v}_{S \setminus \{w_t\}}\big) \right) \tag{1}$$

where $w_t$ is the target word and, $\boldsymbol{V}$ and $\boldsymbol{U}$ are the source n-gram and target word embedding matrices respectively. Here, the set of negative words $N_{w_t}$ is sampled from a multinomial distribution where the probability of picking a word is directly proportional to the square root of its frequency in the corpus. Each target word $w_t$ is sampled with probability $min\{1, \sqrt{t/f_{w_t}} + t/f_{w_t}\}$ where $f_{w_t}$ is the frequency of the word in the corpus.

We adapt the SENT2VEC model to bilingual corpora by introducing a cross-lingual loss in addition to the monolingual loss in equation (1). Given a sentence pair $S = (S_{l_1}, S_{l_2})$ where $S_{l_1}$ and $S_{l_2}$ are translations of each other in languages $l_1$ and $l_2$, the cross-lingual loss for a target word $w_t$ in $l_1$ is given by

$$\ell\big(\boldsymbol{u}_{w_t}^\top \boldsymbol{v}_{S_{l_2}}\big) + \sum_{w' \in N_{w_t}} \ell\big(-\boldsymbol{u}_{w_t'}^\top \boldsymbol{v}_{S_{l_2}}\big) \tag{2}$$

Thus, we use the sentence $S_{l_1}$ to predict the constituent words of $S_{l_2}$ and vice-versa in a similar fashion to the monolingual SENT2VEC, shown in Figure 1. This ensures that the word and n-gram embeddings of both languages lie in the same space.

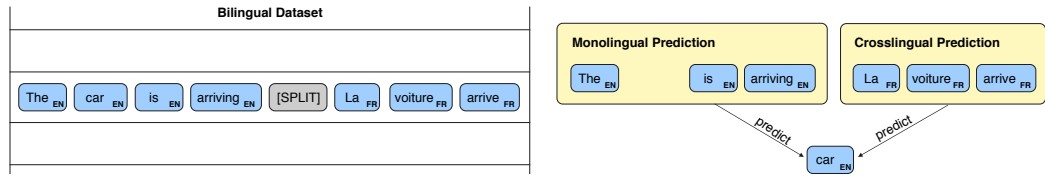

Figure 1: An illustration of the BI-SENT2VEC training process. A word from a sentence pair is chosen as a target and the algorithm learns to predict it using the rest of the sentence(monolingual training component) and the translation of the sentence(cross-lingual component).

Assuming $\boldsymbol{C}$ to be a sentence aligned bilingual corpus and combining equations (1) and (2), our BI-SENT2VEC model objective function is formulated as

$$
\min_{\boldsymbol{U},\boldsymbol{V}} \sum_{\substack{S \in \boldsymbol{C} \\ l,l' \in \{l_1,l_2\} \\ l \neq l'}} \sum_{w_t \in S_l} \left( \underbrace{\ell\big(\boldsymbol{u}_{w_t}^\top \boldsymbol{v}_{S_l \setminus \{w_t\}}\big) + \sum_{w' \in N_{w_t}} \ell\big(-\boldsymbol{u}_{w'}^\top \boldsymbol{v}_{S_l \setminus \{w_t\}}\big)}_{\text{monolingual loss}} + \underbrace{\ell\big(\boldsymbol{u}_{w_t}^\top \boldsymbol{v}_{S_{l'}}\big) + \sum_{w' \in N_{w_t}} \ell\big(-\boldsymbol{u}_{w'}^\top \boldsymbol{v}_{S_{l'}}\big)}_{\text{cross-lingual loss}} \right)
$$
(3)

**Implementation Details.** We build our C++ implementation on the top of the FASTTEXT library (Bojanowski et al., 2016; Joulin et al., 2016). Model parameters are updated by asynchronous SGD with a linearly decaying learning rate.

Our model is trained on the ParaCrawl (Esplà-Gomis, 2019) v4.0 datasets for the English-Italian, English-German, English-French, English-Spanish, English-Hungarian and English-Finnish language pairs. For the English-Russian language pair, we concatenate the OpenSubtitle corpus[1](Lison & Tiedemann, 2016) and the Tanzil project[2] (Quran translations) corpus. The number of parallel sentence pairs in the corpora except for those of English-Finnish and English-Hungarian used by us range from 17-32 Million. Number of parallel sentence pairs for the dis-similar language pairs(English-Hungarian and English-Finnish) is approximately 2 million. Evaluation results for these two language pairs can be found in Subsection 4.4. Exact statistics regarding the different corpora can be found in the Table 7 in the Appendix. All the sentences were tokenized using Spacy tokenizers[3] for their respective languages.

For each dataset, we trained two different models: one with unigram embeddings only, and the other additionally augmented with bigrams. The earlier TRANSGRAM models (Coulmance et al., 2015) were trained on a small amount of data (Europarl Corpus (Koehn, 2005)). To facilitate a fair comparison, we train new TRANSGRAM embeddings on the same data used for BI-SENT2VEC. Given that TRANSGRAM and BI-SENT2VEC are a cross-lingual extension of Skipgram and SENT2VEC respectively, we use the same parameters as (Bojanowski et al., 2016) and (Gupta et al., 2019), except increasing the number of epochs for TRANSGRAM to 8, and decreasing the same for BI-SENT2VEC to 5. Additionally, a preliminary hyperparameter search (except changing the number of epochs) on BI-SENT2VEC and TRANSGRAM did not improve the results. All parameters for training the TRANSGRAM and BI-SENT2VEC models can be found in the Table 6 in the Appendix.

In order to make the comparison more extensive, we also train VECMAP (mapping-based) (Artetxe et al., 2018b;a) and BIVEC (joint-training)(Luong et al., 2015) methods on the same corpora using the exact pipeline as (Ormazabal et al., 2019).

## 4 EVALUATION

To assess the quality of the word and sentence embeddings obtained as well as their cross-lingual alignment quality, we compare our results using the following four benchmarks

- Cross-lingual word retrieval
- Monolingual word representation quality

---

[1] http://www.opensubtitles.org/
[2] http://tanzil.net/
[3] https://spacy.io/

- Cross-lingual sentence retrieval
- Zero-shot cross-lingual transfer of document classifiers

where benchmarks are presented in order of increasing linguistic granularity, i.e. word, sentence, and document level. We also analyze the effect of training data by studying the relationship between representation quality and corpus size.

We use the code available in the MUSE library[4] (Conneau et al., 2017) for all evaluations except the zero-shot classifier transfer, which is tested on the MLDoc task (Schwenk & Li, 2018)[5].

## 4.1 WORD TRANSLATION

The task involves retrieving correct translation(s) of a word in a source language from a target language. To evaluate translation accuracy, we use the bilingual dictionaries constructed by (Conneau et al., 2017). We consider 1500 source-test queries and 200k target words for each language pair and report P@1 scores for the supervised and unsupervised baselines as well as our models in Table 1.

| Method | en-es | | en-fr | | en-de | | en-ru | | en-it | | avg. |
|---|---|---|---|---|---|---|---|---|---|---|---|
| | → | ← | → | ← | → | ← | → | ← | → | ← | |
| MUSE (Conneau et al., 2017) | 81.7 | 83.3 | 82.3 | 82.1 | 74.0 | 72.2 | 44.0 | 59.1 | 78.6 | 77.9 | 73.5 |
| UMWE (Chen & Cardie, 2018) | 82.5 | 83.1 | 82.5 | 82.1 | 74.6 | 72.5 | 49.5 | 61.7 | 78.3 | 77.0 | 74.4 |
| Procrustes + refine (Conneau et al., 2017) | 82.4 | 83.9 | 82.3 | 83.2 | 75.3 | 73.2 | 50.1 | 63.5 | 77.5 | 77.6 | 74.9 |
| RCSLS (Joulin et al., 2018) | 83.7 | 87.1 | 84.1 | 84.7 | 79.2 | 77.5 | 60.9 | 70.2 | 81.1 | 82.7 | 79.1 |
| TRANSGRAM (Coulmance et al., 2015) | **91.6** | 88.6 | 89.1 | 90.1 | **87.5** | 87.2 | **65.6** | **73.7** | 88.6 | 89.5 | 85.2 |
| VECMAP (unsupervised) (Artetxe et al., 2018b) | 87.4 | 87.8 | 88.3 | 88.5 | 84.3 | 87.2 | 48.6 | 50.5 | 87.4 | 86.5 | 79.6 |
| VECMAP (supervised) (Artetxe et al., 2018a) | 87.2 | 90.2 | 87.6 | 90.4 | 87.3 | 86.8 | 49.7 | 65.6 | 87.2 | 89.2 | 82.1 |
| BIVEC NN (Luong et al., 2015) | 87.4 | 88.6 | 86.8 | 89.1 | **87.5** | 87.2 | 64.0 | 59.1 | 86.8 | 84.0 | 81.7 |
| BIVEC CSLS (Luong et al., 2015) | 87.6 | 89.1 | 88.8 | 90.3 | 86.4 | 87.2 | 66.1 | 70.6 | 87.6 | 87.8 | 84.3 |
| BI-SENT2VEC uni. NN | 86.9 | 91.6 | 86.9 | 91.0 | 86.0 | 88.7 | 58.0 | 72.8 | 88.3 | **92.4** | 84.3 |
| BI-SENT2VEC uni. + bi. NN | 89.4 | **92.9** | **89.3** | 92.8 | 86.7 | **89.3** | 59.0 | 70.2 | 89.5 | 91.8 | 85.1 |
| BI-SENT2VEC uni. CSLS | 86.0 | 91.7 | 86.4 | 91.4 | 84.6 | 88.8 | 60.5 | 73.0 | 88.2 | 91.8 | 84.2 |
| BI-SENT2VEC uni. + bi. CSLS | 89.0 | 92.1 | 88.9 | 92.4 | 86.5 | 89.0 | 61.0 | 73.5 | **89.6** | 91.4 | **85.3** |

Table 1: **Word translation retrieval P@1 for various language pairs of MUSE evaluation dictionary** (Conneau et al., 2017). NN: nearest neighbours. CSLS: Cross-Domain Similarity Local Scaling. ('en' is English, 'fr' is French, 'de' is German, 'ru' is Russian, 'it' is Italian) ('uni.' and 'bi.' denote unigrams and bigrams respectively) (→ denotes translation from the first language to the second and ← the other way around.)

## 4.2 MONOLINGUAL WORD REPRESENTATION QUALITY

We assess the monolingual quality improvement of our proposed cross-lingual training by evaluating performance on monolingual word similarity tasks. To disentangle the specific contribution of the cross-lingual loss, we train the monolingual counterpart of BI-SENT2VEC, SENT2VEC on the same corpora as our method.

Performance on monolingual word-similarity tasks is evaluated using the *English SimLex-999* (Hill et al., 2014) and its Italian and German translations, *English WS-353* (Finkelstein et al., 2001) and its German, Italian and Spanish translations. For French, we use a translation of the *RG-65* (Joubarne & Inkpen, 2011) dataset. Pearson scores are used to measure the correlation between human-annotated word similarities and predicted cosine similarities. We also include FASTTEXT monolingual vectors trained on CommonCrawl data (Grave et al., 2018a) which is comprised of 600 billion, 68 billion, 66 billion, 72 billion and 36 billion words of English, French, German, Spanish and Italian respectively and is at least 100× larger than the corpora on which we trained BI-SENT2VEC. We report Pearson correlation scores on different word-similarity datasets for En-It pair in Table 2. Evaluation results on other language pairs are similar and can be found in the appendix in Tables 8, 9, and 10.

---

[4]https://github.com/facebookresearch/MUSE
[5]https://github.com/facebookresearch/MLDoc

| Method\Dataset | SimLex-999 | | WS-353 | |
|---|---|---|---|---|
| | en | it | en | it |
| MUSE | 0.38 | 0.30 | 0.74 | 0.64 |
| RCSLS | 0.38 | 0.30 | 0.74 | 0.64 |
| FASTTEXT- Common Crawl | 0.49 | 0.32 | 0.75 | 0.57 |
| BIVEC | 0.40 | 0.36 | 0.70 | 0.60 |
| TRANSGRAM | 0.43 | 0.37 | 0.73 | 0.63 |
| SENT2VEC uni. | 0.49 | 0.38 | 0.73 | 0.60 |
| BI-SENT2VEC uni. | 0.57 | 0.47 | 0.79 | 0.65 |
| BI-SENT2VEC uni. + bi. | **0.58** | **0.50** | **0.80** | **0.69** |

Table 2: **Monolingual word similarity task performance of our methods when trained on en-it ParaCrawl data.** We report Pearson correlation scores.

## 4.3 CROSS-LINGUAL SENTENCE RETRIEVAL

The primary contribution of our work is to deliver improved cross-lingual sentence representations. We test sentence embeddings for each method obtained by bag-of-words composition for sentence retrieval across different languages on the Europarl corpus. In particular, the tf-idf weighted average is used to construct sentence embeddings from word embeddings. We consider 2000 sentences in the source language dataset and retrieve their translation among 200K sentences in the target language dataset. The other 300K sentences in the Europarl corpus are used to calculate tf-idf weights. Results for P@1 of unsupervised and supervised benchmarks vs our models are included in Table 3.

| Method | en-es | | en-fr | | en-de | | en-it | | avg. |
|---|---|---|---|---|---|---|---|---|---|
| | $\rightarrow$ | $\leftarrow$ | $\rightarrow$ | $\leftarrow$ | $\rightarrow$ | $\leftarrow$ | $\rightarrow$ | $\leftarrow$ | |
| MUSE | 72.7 | 71.5 | 69.2 | 68.8 | 53.3 | 53.4 | 66.1 | 64.3 | 64.9 |
| RCSLS | 26.9 | 26.7 | 19.3 | 21.2 | 8.8 | 11.3 | 15.1 | 17.6 | 18.4 |
| TRANSGRAM | 83.5 | 81.4 | 80.4 | 81.6 | 64.8 | 69.9 | 77.2 | 77.9 | 77.1 |
| VECMAP (unsupervised) | 81.7 | 82.1 | 79.8 | 80.4 | 62.8 | 64.6 | 69.0 | 71.1 | 74.0 |
| VECMAP (supervised) | 81.3 | 81 | 80.4 | 80.7 | 62.6 | 64.3 | 67.8 | 71 | 73.6 |
| BIVEC NN | 69.8 | 77.1 | 54.7 | 75.5 | 56.1 | 44.1 | 58.2 | 45.1 | 60.1 |
| BIVEC CSLS | 81.6 | 83.4 | 78.1 | 81.6 | 71.6 | 68.1 | 74.2 | 72.4 | 76.4 |
| BI-SENT2VEC uni. NN | 87.8 | 86.4 | 85.2 | 83.4 | 82.3 | 80.2 | 85.9 | 85.8 | 84.6 |
| BI-SENT2VEC uni. + bi. NN | 87.9 | 87.8 | 86.1 | 83.9 | 79.5 | 79.7 | 85.1 | 85.3 | 84.4 |
| BI-SENT2VEC uni. CSLS | 89.5 | 88.5 | 87.1 | 86.4 | **84.4** | 83.0 | **88.2** | 87.5 | 86.8 |
| BI-SENT2VEC uni. + bi. CSLS | **89.7** | **89.6** | **87.8** | **87.4** | 84.2 | **84.0** | 87.9 | **87.6** | **87.3** |
| Reduction in error | 37.5% | 37.3% | 37.8% | 31.5% | 44.4% | 46.8% | 46.9% | 43.9% | – |

Table 3: **Cross-lingual Sentence retrieval.** We report P@1 scores for 2000 source queries searching over 200 000 target sentences. Reduction in error is calculated with respect to BI-SENT2VEC uni. + bi. CSLS and the best non-BI-SENT2VEC method.

## 4.4 PERFORMANCE ON DIS-SIMILAR LANGUAGE PAIRS

We report a substantial improvement on the performance of previous models on cross-lingual word and sentence retrieval tasks for the dis-similar language pairs(English-Finnish and English-Hungarian). We use the same evaluation scheme as in Subsections 4.1 and 4.3 Results for these pairs are included in Table 4.

## 4.5 ZERO-SHOT CROSS-LINGUAL TRANSFER OF DOCUMENT CLASSIFIERS

The MLDoc multilingual document classification task (Schwenk & Li, 2018) consists of news documents given in 8 different languages, which need to be classified into 4 different categories. To demonstrate the ability to transfer trained classifiers in a robust fashion between languages, we

| Method | word retrieval | | | | sentence retrieval | | | |
| --- | --- | --- | --- | --- | --- | --- | --- | --- |
| | en-fi | | en-hu | | en-fi | | en-hu | |
| | → | ← | → | ← | → | ← | → | ← |
| MUSE | 48.1 | 59.5 | 53.9 | 64.9 | 21.7 | 29.5 | 39.1 | 46.7 |
| RCSLS | 61.8 | 69.9 | 67.0 | 73.0 | 3.2 | 4.8 | 3.6 | 5.1 |
| VECMAP (unsupervised) | 62.5 | 66.8 | 61.6 | 68.7 | 13.2 | 14.7 | 20.5 | 19.3 |
| VECMAP (supervised) | 62.6 | 78.3 | 63.7 | 76.6 | 15.0 | 16.9 | 20.9 | 21.7 |
| BIVEC NN | 62.1 | 55.3 | 62.1 | 53.7 | 14.2 | 9.7 | 26.2 | 13.7 |
| BIVEC CSLS | 69.6 | 78.0 | 72.4 | 78.4 | 33.3 | 32.0 | 46.7 | 41.3 |
| TRANSGRAM | 69.7 | 81.1 | 73.1 | 80.8 | 35.4 | 40.5 | 52.1 | 55 |
| BI-SENT2VEC uni. NN | 71.2 | 85.4 | 75.6 | 83.9 | 63.5 | 64.2 | 75.2 | 76.2 |
| BI-SENT2VEC uni. + bi. NN | 68.5 | 81.7 | 71.4 | 79.4 | 57.5 | 55.9 | 65.8 | 65.2 |
| BI-SENT2VEC uni. CSLS | **72.0** | **86.5** | **76.3** | **85.1** | **70.2** | **69.0** | **81.4** | **80.8** |
| BI-SENT2VEC uni. + bi. CSLS | 70.1 | 84.4 | 73.7 | 81.7 | 66 | 64.1 | 73.8 | 74.5 |

Table 4: **Cross-lingual Word and Sentence retrieval for dis-similar language pairs (P@1 scores).** 'en' is English, 'fi' is Finnish, 'hu' is Hungarian

use a zero-shot setting, i.e., we train a classifier on embeddings in the source language, and report the accuracy of the same classifier applied to the target language. As the classifier, we use a simple feed-forward neural network with two hidden layers of size 10 and 8 respectively, optimized using the Adam optimizer. Each document is represented using the sum of its sentence embeddings.

| Method | en-es | | en-fr | | en-de | | en-it | | avg. |
| --- | --- | --- | --- | --- | --- | --- | --- | --- | --- |
| | → | ← | → | ← | → | ← | → | ← | |
| LASER | **79.3** | 69.6 | **78.0** | 80.1 | 86.3 | **80.8** | 70.2 | **74.2** | 77.3 |
| BI-SENT2VEC | 74.0 | **71.5** | 81.6 | **82.2** | **86.5** | 79.2 | **75.0** | 72.6 | **77.8** |

Table 5: MLDoc Benchmark results (Schwenk & Li, 2018). A document classifier was trained on one language and tested on another without additional training/fine-tuning. We report % accuracy.

We compare the performance of BI-SENT2VEC with the LASER sentence embeddings (Artetxe & Schwenk, 2018) in Table 5. LASER sentence embedding model is a multi-lingual sentence embedding model which is composed of a biLSTM encoder and an LSTM decoder. It uses a shared byte pair encoding based vocabulary of 50k words. The LASER model which we compare to was trained on 223M sentences for 93 languages and requires 5 days to train on 16 V100 GPUs compared to our model which takes 1-2.5 hours for each language pair on 30 CPU threads.

### 4.6 EFFECT OF CORPUS SIZE ON REPRESENTATION QUALITY

We conduct an ablation study on how BI-SENT2VEC embeddings' performance depends on the size of the training corpus. We uniformly sample smaller subsets of the En-Fr ParaCrawl dataset and train a BI-SENT2VEC model on them. We test word/sentence translation performance with the CSLS retrieval criterion, and monolingual embedding quality for En-Fr with increasing ParaCrawl corpus size. The results are illustrated in Figures 2 and 3.

## 5 DISCUSSION

In the following section, we discuss the results on monolingual and cross-lingual benchmarks, presented in Tables 1 - 5, and a data ablation study for how the model behaves with increasing parallel corpus size in Figure 2 - 3. The most impressive outcome of our experiments is improved cross-lingual sentence retrieval performance, which we elaborate on along with word translation in the next subsection.

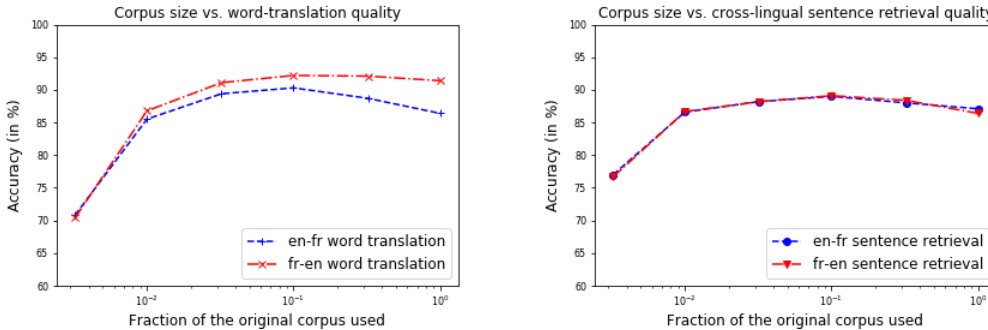

Figure 2: **Effect of corpus size on cross-lingual word/sentence retrieval performance.**

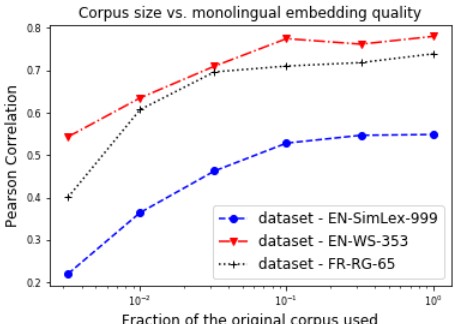

Figure 3: **Effect of corpus size on monolingual word quality.** We use SimLex-999, WS-353, and FR-RG datasets for measuring monolingual word embedding quality.

**Cross-lingual evaluations**     For cross-lingual tasks, we observe in Table 1 that jointly trained embeddings produce much better results on cross-lingual word and sentence retrieval tasks. BI-SENT2VEC's performance on word-retrieval tasks is uniformly superior to mapping methods, achieving up to 11.5% more in P@1 than RCSLS for the English to German language pair, consistent with the results from (Ormazabal et al., 2019). It is also on-par with, or better than competing joint methods except on translation from Russian to English, where TRANSGRAM receives a significantly better score. For word retrieval tasks, there is no discernible difference between CSLS/NN criteria for BI-SENT2VEC, suggesting the relative absence of the hubness phenomenon which significantly hinders the performance of cross-lingual word embedding methods.

Our principal contribution is in improving cross-lingual sentence retrieval. Table 3 shows BI-SENT2VEC decisively outperforms all other methods by a wide margin, reducing the relative P@1 error anywhere from 31.5% to 55.1%. Our model displays considerably less variance than others in quality across language pairs, with at most a $\approx 5\%$ deficit between best and worst, and nearly symmetric accuracy within a language pair.

TRANSGRAM also outperforms the mapping-based methods, but still falls significantly short of BI-SENT2VEC's. These results can be attributed to the fact that BI-SENT2VEC directly optimizes for obtaining robust sentence embeddings using additive composition of its word embeddings. Since BI-SENT2VEC's learning objective is closest to a sentence retrieval task amongst current state-of-the-art methods, it can surpass them without sacrificing performance on other tasks.

**Cross-lingual evaluations on dis-similar language pairs**     Unlike other language pairs in the evaluation, English-Finnish and English-Hungarian pairs are composed of languages from two different language families(English being an Indo-European language and the other language being a Finno-Ugric language). In Table 4, we see that the performance boost achieved by BI-SENT2VEC on competing methods methods is more pronounced in the case of dis-similar language pairs as compared to paris of languages close to each other. This observation affirms the suitability of BI-SENT2VEC for learning joint representations on languages from different families.

**Monolingual word quality**    For the monolingual word similarity tasks, we observe large gains over existing methods. SENT2VEC is trained on the same corpora as us, and FASTTEXT vectors are trained on the CommonCrawl corpora which are more than 100 times larger than ParaCrawl v4.0. In Table 2, we see that BI-SENT2VEC outperforms them by a significant margin on SimLex-999 and WS-353, two important monolingual word quality benchmarks. This observation is in accordance with the fact (Faruqui & Dyer, 2014) that bilingual contexts can be surprisingly effective for learning monolingual word representations. However, amongst the joint-training methods, BI-SENT2VEC also outperforms TRANSGRAM and BIVEC trained on the same corpora by a significant margin, again hinting at the superiority of the sentence level loss function over a fixed context window loss.

**Effect of n-grams**    (Gupta et al., 2019) report improved results on monolingual word representation evaluation tasks for SENT2VEC and FASTTEXT word vectors by training them alongside word n-grams. Our method incorporates their results based on the observation that unigram vectors trained alongside with bigrams significantly outperform unigrams alone on the majority of the evaluation tasks. We can see from Tables 1 - 3 that this holds for the bilingual case as well. However, in case of dis-similar language pairs(Table 4), we observe that using n-grams degrades the cross-lingual performance of the embeddings. This observation suggests that use of higher order n-grams may not be helpful for language pairs where the grammatical structures are contrasting.

**Effect of corpus size**    Considering the cross-lingual performance curve exhibited by BI-SENT2VEC in Figure 2, increasing corpus size for the English-French datasets up to 1-3.1M lines appears to saturate the performance of the model on cross-lingual word/sentence retrieval, after which it either plateaus or degrades slightly. This is an encouraging result, indicating that joint methods can use significantly less data to obtain promising performance. This implies that joint methods may not necessarily be constrained to high-resource language pairs as previously assumed, though further experimentation is needed to verify this claim.

It should be noted from Figure 3 that the monolingual quality does keep improving with an increase in the size of the corpus. A potential way to overcome this issue of plateauing cross-lingual performance is to give different weights to the monolingual and cross-lingual component of the loss with the weights possibly being dependent on other factors such as training progress.

**Comparison with a cross-lingual sentence embedding model and performance on document level task**    On the MLDoc classifier transfer task (Schwenk & Li, 2018) where we evaluate a classifier learned on documents in one language on documents in another, Table 5 shows we achieve parity with the performance of the LASER model for language pairs involving English, where BI-SENT2VEC's average accuracy of 77.8% is slightly higher than LASER's 77.3%. While the comparison is not completely justified as LASER is multilingual in nature and is trained on a different dataset, one must emphasize that BI-SENT2VEC is a bag-of-words method as compared to LASER which uses a multi-layered biLSTM sentence encoder. Our method only requires to average a set of vectors to encode sentences reducing its computational footprint significantly. This makes BI-SENT2VEC an ideal candidate for on-device computationally efficient cross-lingual NLP, unlike LASER which has a huge computational overhead and specialized hardware requirement for encoding sentences.

# 6    CONCLUSION AND FUTURE WORK

We introduce a cross-lingual extension of an existing monolingual word and sentence embedding method. The proposed model is tested at three levels of linguistic granularity: words, sentences and documents. The model outperforms all other methods by a wide margin on the cross-lingual sentence retrieval task while maintaining parity with the best-performing methods on word translation tasks. Our method achieves parity with LASER on zero-shot document classification, despite being a much simpler model. We also demonstrate that training on parallel data yields a significant improvement in the monolingual word representation quality.

The success of our model on the bilingual level calls for its extension to the multilingual level especially for pairs which have little or no parallel corpora. While the amount of bilingual/multilingual parallel data has grown in abundance, the amount of monolingual data available is practically limitless. Consequently, we would like to explore training cross-lingual embeddings with a large amount of raw text combined with a smaller amount of parallel data.

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

# A APPENDIX

## A.1 DATASET STATISTICS

| Dataset | Number of sentences | Number of tokens (English tokens if bilingual) |
|---|---|---|
| En-De ParaCrawl v4.0 | 17 Million | 308 Million |
| En-Es ParaCrawl v4.0 | 22 Million | 477 Million |
| En-Fi ParaCrawl v4.0 | 2.16 Million | 42 Million |
| En-Fr ParaCrawl v4.0 | 32 Million | 665 Million |
| En-Hu ParaCrawl v4.0 | 1.91 Million | 31 Million |
| En-It ParaCrawl v4.0 | 13 Million | 261 Million |
| En-Ru OpenSubtitles + Tanzil | 27 Million | 363 Million |
| Wikipedia - En | 70 Million | 1792 Million |
| Wikipedia - De | – | 1384 Million |
| Wikipedia - Fr | – | 1108 Million |
| Wikipedia - Es | – | 797 Million |
| Wikipedia - It | – | 702 Million |
| Wikipedia - Ru | – | 824 Million |
| Common Crawl - En | – | 600 Billion |
| Common Crawl - De | – | 66 Billion |
| Common Crawl - Fr | – | 68 Billion |
| Common Crawl - It | – | 36 Billion |
| Common Crawl - Es | – | 72 Billion |

Table 6: **Dataset Sizes**. 'En','De','Fi','Fr','Hu','It','Es' and 'Ru' stand for English, German, Finnish, French, Hungarian, Italian, Spanish and Russian respectively.

We used ParaCrawl v4.0 corpora for training BI-SENT2VEC, SENT2VEC,BIVEC,VECMAP and TRANSGRAM embeddings except for En-Ru pair for which we used OpenSubtitles and Tanzil corpora combined. MUSE and RCSLS vectors were trained from FASTTEXT vectors obtained from Wikipedia dumps(Grave et al., 2018a).

## A.2 TRAINING PARAMETERS FOR TRAINED MODELS

| Model | BI-SENT2VEC uni. | BI-SENT2VEC uni. + bi. | SENT2VEC uni. | TRANSGRAM |
|---|---|---|---|---|
| Embedding dimension | 300 | 300 | 300 | 300 |
| Max vocabulary size | 750k | 750k | 750k | 750k |
| Minimum word count | 5 | 8 | 5 | 5 |
| Initial Learning Rate | 0.2 | 0.2 | 0.2 | 0.025 |
| Epochs | 5 | 5 | 5 | 8 |
| Subsampling hyper-parameter | $1 \cdot 10^{-5}$ | $5 \cdot 10^{-6}$ | $1 \cdot 10^{-5}$ | $1 \cdot 10^{-4}$ |
| Word-Ngrams Bucket Size | – | 2M | – | – |
| Word-Ngrams dropped per context | – | 4 | – | – |
| Window size | | | | 5 |
| Number of negatives sampled | 10 | 10 | 10 | 5 |

Table 7: Hyperparameters for the trained models

## A.3    ADDITIONAL MONOLINGUAL QUALITY TABLES

| Method\Dataset | SimLex-999 en | WS-353 en | es |
|---|---|---|---|
| MUSE | 0.38 | 0.74 | 0.61 |
| RCSLS | 0.38 | 0.74 | 0.62 |
| FASTTEXT- Common Crawl | 0.49 | 0.75 | 0.54 |
| BIVEC | 0.40 | 0.72 | 0.57 |
| TRANSGRAM | 0.42 | 0.74 | 0.59 |
| SENT2VEC uni. | 0.49 | 0.58 | 0.51 |
| BI-SENT2VEC uni. | 0.57 | 0.78 | 0.60 |
| BI-SENT2VEC uni. + bi. | **0.60** | **0.82** | **0.66** |

Table 8: **Monolingual word similarity task performance of our methods when trained on en-es ParaCrawl data.** We report Pearson correlation scores.

| Method\Dataset | SimLex-999 en | WS-353 en | RG-65 fr |
|---|---|---|---|
| MUSE | 0.38 | 0.74 | 0.72 |
| RCSLS | 0.38 | 0.74 | 0.70 |
| FASTTEXT- Common Crawl | 0.49 | 0.75 | 0.76 |
| BIVEC | 0.40 | 0.70 | 0.74 |
| TRANSGRAM | 0.39 | 0.72 | 0.74 |
| SENT2VEC uni. | 0.46 | 0.75 | 0.71 |
| BI-SENT2VEC uni. | 0.55 | 0.78 | 0.74 |
| BI-SENT2VEC uni. + bi. | **0.59** | **0.79** | **0.78** |

Table 9: **Monolingual word similarity task performance of our methods when trained on en-fr ParaCrawl data.** We report Pearson correlation scores.

| Method\Dataset | SimLex-999 en | de | WS-353 en | de |
|---|---|---|---|---|
| MUSE | 0.38 | 0.41 | 0.74 | 0.68 |
| RCSLS | 0.38 | 0.43 | 0.74 | **0.70** |
| FASTTEXT- Common Crawl | 0.49 | 0.39 | 0.75 | 0.64 |
| BIVEC | 0.40 | 0.41 | 0.71 | 0.62 |
| TRANSGRAM | 0.42 | 0.42 | 0.74 | 0.66 |
| SENT2VEC uni. | 0.48 | 0.38 | 0.70 | 0.63 |
| BI-SENT2VEC uni. | 0.56 | 0.47 | **0.76** | 0.68 |
| BI-SENT2VEC uni. + bi. | **0.59** | **0.53** | 0.75 | **0.70** |

Table 10: **Monolingual word similarity task performance of our methods when trained on en-de ParaCrawl data.** We report Pearson correlation scores.

