# OpenReview forum: "Robust Cross-lingual Embeddings from Parallel Sentences "
_ICLR.cc/2020/Conference — Reject_

### Official Review · AnonReviewer2 · 2019-10-21
**Official Blind Review #2**

**Rating:** 3

**Review:**


***Update***

I thank the reviewers for answering my questions, and I have read the reviews from the other reviewers.  I am borderline on this paper, but still learn towards rejection. I feel that it is a bit incremental and still a little misleading/over-stated. For instance, xnli isn't mentioned but mldoc is, it isn't clear in the table which methods are mappings and which aren't (big data difference), in the mldoc experiment was more parallel data used for these specific languages than was used for LASER? the experiment I asked for wasn't done on the same corpora head-to-head with LASER and their approach. I think any claims about outperforming LASER need to be evaluated on the same corpus as much as possible (Europarl) and weaknesses of the approach should also be mentioned (it is not multilingual in these evaluations). Otherwise you conflate data/multilingualness/model which makes it hard to draw conclusions from the experiments. Also CSLS  was not applied to the ACL paper and this makes a huge difference and should be accounted for (and also idf possibly). Also this paper still isn't at least mentioned which is very related in my opinion.

This paper proposes a method to learn bilingual word embeddings by modifying the Sent2Vec (which is based on word2vec) approach, and applying it to bitext. They evaluate on monolingual and bilingual word similarity, bitext mining, and a zero-shot document classification task where a classifier is trained on data in one language but evaluated on another (using their embeddings as features in both cases).

I have the following concerns about this paper:

1) For the sentence mining tasks - how come Ormazabal 2019 is not included? The baselines are weak and the task is not standard. One of the baselines used here is MUSE which is unsupervised (The refined version of MUSE was also not included in these experiments). There are fixed datasets that people have experimented with (like BUCC) that would help tie in your results here with the literature a little better. These could include LASER (which is already compared to for document classification) and "Simple and Effective Paraphrastic Similarity from Parallel Translations" ACL 2019, which like this paper, proposes a pooled token embedding approach and outperforms more complicated architectures. Neither of these approaches uses idf either (how much does idf help - is it really needed?).

2) For the zero-shot document classification task, it should also be pointed out that LASER can handle many languages all at once. Can this approach also work well if all languages were trained jointly?. Also did you evaluate on XNLI for zero-shot as well? LASER does very well here. I do realize comparing to LASER is not really fair since it is trained on so much data, however the model is similar to previous versions of LASER that were trained on Europarl, which could also be used as the data for training your models for a more apples-to-apples comparison.

3) Note that also the improvements on monolingual similarity using parallel data are well known. For instance "Embedding Word Similarity with Neural Machine Translation" (arXiv 2014).. Also even the base for the current state of the art models on SimLex-999, Paragram (TACL 2015), used paraphrases created by pivoting on parallel data.

I think the main contributions of this paper are modifying Sent2vec so it can be used on bilingual data and using it to learn nice representations for words and sentences (and documents). I think that to be published, it should clearly outperform and/or have advantages over all previous works - and this is not clear from this paper in its current form. It is okay if it doesn't do the best on everything, but it is hard to tell how this work currently fits into the literature especially in terms of the sentence-level tasks which are a focus.

**Experience Assessment:**

I have published in this field for several years.

**Review Assessment: Checking Correctness Of Derivations And Theory:**

N/A

**Review Assessment: Checking Correctness Of Experiments:**

I carefully checked the experiments.

**Review Assessment: Thoroughness In Paper Reading:**

N/A

---

> ### Author Response · Authors · 2019-11-15
> **Answer to Official Blind Review #2**
>
> We thank the reviewer for the feedback and comments.
>
> 1)
> We thank you for your suggestion of including the BiVec results from Ormazbal et al. We trained BiVec models using the same pipeline as Ormazabal et al. on our datasets. We found out that in most of the cases BiVec models performs poorly, or on par with TransGram (cross-lingual word retrieval, cross-lingual sentence retrieval and monolingual word quality tasks) and consistently lags behind Bi-Sent2Vec by a significant margin (see Table 1-4) . We have also added VecMap (Table 1-4) as a stronger mapping-based method baseline in the comparison.
>
>
> Regarding the usage of tf-idf for sentence retrieval: not using tf-idf leads to a significant drop in the performance of all three mapping based methods as well as jointly trained methods used in the comparison.
>
> 2)
> In the paper, we point out that during preliminary experiments to train multilingual parallel data, we saw a drop in performance of our models and as a result include obtaining better embeddings from multilingual data in place of bilingual data in the future work section. Due to time constraints, we were unable to finish running other experiments. We will add experimental results comparing to other models as well.
>
> 3)
> We acknowledge that our claim that multilingual training can improve performance on monolingual tasks has already been shown before to hold true and we have cited the relevant papers in the revised version. However, we also demonstrate significant improvement in monolingual quality for our method compared to other bilingually trained methods.
>
>
> In order to strengthen our claims further, we trained Bi-Sent2Vec, TransGram and BiVec on the English - Finnish and English - Hungarian ParaCrawl corpora v4 (the languages in these pairs come from different language families). The size of both corpora was around 2M sentences and hence much smaller in comparison to other corpora used in the experiments. We obtained superior results (see Subsection 4.4 and Table 4 of the revised paper) to Bivec as well as TransGram on both cross-lingual word and sentence retrieval tasks. In fact, Bi-Sent2Vec showed greater improvements for these 2 language pairs as compared to high resource language pairs indicating another usefulness of Bi-Sent2Vec.
>
> Moreover, it is important to note that we show a significant improvement on the previous joint training methods except cross-lingual word translation on similar language pairs where we are on-par with the existing methods. On dis-similar language pairs, we outperform these methods by an even larger margin.
>
> Our similar performance  to LASER on MLDoc was added to show another advantage of our model in addition to being computationally cheaper.
>
> Lastly, we would like the reviewer to note that our model is a bag-of-words in the spirit of previous joint-training models and it outperforms them in all the departments except on cross-lingual word translations where it is on-par with the competing methods. Moreover, on dis-similar language pairs, we report superiority on all the competing models on the cross-lingual word translations as well and the improvement on the competing models is more pronounced. We have added extra language-pairs and new competing methods to make the comparison more extensive.
>
> We thank the reviewer for their recommendations which have helped improve the paper.

---

### Official Review · AnonReviewer3 · 2019-10-22
**Official Blind Review #3**

**Rating:** 3

**Review:**

The paper does not bring anything novel to the field of cross-lingual representation learning: it just revisits some older ideas (from the period of 2013-2015), now revamped, given the fact that more sophisticated and more effective methods are used to model exactly the same intuitions. I see this work as largely incremental, and it just further supports what has been known before, and it further supports recent findings (which are all quite straightforward) from the work of Ormazabal et al. (ACL 2019). The actual model implementation is a straightforward extension of the Sent2Vec model to cross-lingual scenarios, inspired by previous work (e.g., the work on TransGram and BiVec), so the paper is also very incremental from the methodological perspective.

I am puzzled why MUSE is selected as the unsupervised baseline given that fact that: 1) previous work showed its non-robustness for many language pairs, 2) the VecMap model of Artetxe et al. has been proven as the most robust unsupervised cross-lingual word embedding model in several recent empirical analyses - see e.g., Glavas et al. (ACL 2019), Heyman et al. (NAACL 2019), or the original VecMap work. Also, I am puzzled why the paper overstates the rekindled interest towards TransGram, given that TransGram and especially BiVec are well-known models that learn from parallel data.

Another note related to evaluation: to really establish how different cross-lingual embeddings compare to each other, a wider set of experiments and downstream evaluation is definitely required, see the work of e.g., Glavas et al. (ACL 2019).

Most importantly, the paper evaluates only on very similar language pairs. The main reason why much recent work has focused on alignment-based/projection-based methods was quite pragmatic: we need such weakly supervised methods where we cannot assume the abundance of parallel data to enable cross-lingual transfer in resource-poor settings. If parallel data exists, it is quite intuitive and obvious (and also empirically validated before) that joint modeling is a better choice than a weakly supervised method that just uses 1K or 5K translation pairs. In fact, I am not even sure that it is fair to compare models that rely only on 1K translation pairs with models that draw their strength from 1M or 2M parallel sentences. This paper just shows that, if we have parallel data (which we do for many resource-richer language pairs), it is better to do joint modeling instead of learning simple alignments, but that is a pretty trivial finding imho.

Are the results on MLDoc really state-of-the-art? The results are actually quite mixed, and the advantage of Bi-Sent2Vec is its quicker training. However, what about more recent methods such as XLM which rely on exactly the same resources as Bi-Sent2Vec to do the zero-shot classification task?

Table 2: it is a well-known fact that multilingual training can improve performance in monolingual supervision: see e.g., the work of Faruqui and Dyer (EACL 2014, not cited). Alignment-based approaches that apply the Orthogonal Procrustes mapping cannot improve on monolingual word similarity simply because the orthogonality constraint preserves the topology of the original space. Therefore, evaluating different embedding methods on the intrinsic word similarity task is not a sound evaluation protocol imho - it would be much more informative to plug the embeddings as features in a classification or a parsing task (or something else).

Figure 2: corpus size. Based on the results presented, it seems that the performance saturates by adding more parallel data, but the authors fail to fully understand their evaluation data in the first place. For instance, there are multiple problems with the MUSE datasets, as discussed in the recent work of Kementchedjhieva et al. (EMNLP 2019) - it evaluates mostly high-frequent word (actually - noun) translation, and of course that this saturates more quickly. It doesn't by any means imply that joint training therefore requires less data to reach peak performance: this is true only with the MUSE dataset, and is not a general truth.

Minor:
- The work of Artetxe et al. (ACL 2017) should be cited when talking about bootstrapping alignment-based methods from limited bilingual supervision (instead of the work of Artetxe and Schwenk which concerns learning multilingual sentence embeddings).
- Many very relevant and historically important papers are omitted from the related work section: e.g., Hermann and Blunsom's work, Chandar et al., Soyer et al., Vulic and Moens, Gouws et al., Levy et al., to name only a few.
- I am not sure that the statement that BiVec is not needed in the presence of TransGram is true in general: it mostly suggests that there are some deficiencies with the evaluation protocol.

**Experience Assessment:**

I have published in this field for several years.

**Review Assessment: Checking Correctness Of Derivations And Theory:**

I carefully checked the derivations and theory.

**Review Assessment: Checking Correctness Of Experiments:**

I carefully checked the experiments.

**Review Assessment: Thoroughness In Paper Reading:**

I read the paper at least twice and used my best judgement in assessing the paper.

---

> ### Author Response · Authors · 2019-11-15
> **Answer to Official Blind Review #3**
>
> We thank the reviewer for the feedback and comments.
>
> We have added the missing references wherever possible.
>
> Following your feedback regarding the unsupervised baseline, we performed preliminary experiments on VecMap and the results obtained by both TransGram and Bi-Sent2vec consistently outperform VecMap as well. We have added VecMap as a stronger baseline in the paper. We also trained BiVec models using the same pipeline as Ormazabal et al. to facilitate the comparison. The reviewers can refer to these new evaluations in Tables 1-4 of the updated manuscript. We found that in most cases BiVec models performed poorly or on par with TransGram (on cross-lingual word retrieval,  cross-lingual sentence retrieval and monolingual word quality tasks) strengthening the validity of our claim that TransGram is superior to BiVec models, and training skipgram-like models on word-aligned parallel corpora doesn’t improve performance over similar models on only sentence-aligned parallel corpora.
>
> To address your concern regarding our evaluation of only very similar language pairs, we trained Bi-Sent2Vec, TransGram and BiVec on the English - Finnish and English - Hungarian ParaCrawl corpora v4 (the languages in these pairs come from different language families). The size of both corpora was around 2M sentences and hence much smaller in comparison to other corpora used in the experiments. The results are included in the paper (Section 4.4 - Table 4). We obtained superior performances than BiVec as well as TransGram on both cross-lingual word and sentence retrieval. Moreover, we outperformed both of these methods on the monolingual word quality tasks. In fact, Bi-Sent2Vec showed greater improvements for these 2 language pairs as compared to similar language pairs indicating another usefulness of Bi-Sent2Vec.
>
> We also performed preliminary experiments for the BUCC evaluation task. For fr-en, de-en, and ru-en language pairs, we applied the same margin-based classifier as LASER to candidates retrieved using the CSLS criterion. As shown in Table 4 of the most recent version of the LASER paper, training and test results are similar for all methods, so we expect the same thing for BiSent2Vec. Our results for BUCC support our findings on Europarl sentence retrieval. We share a table summarizing our results (https://imgur.com/0qwBpAD), and will update the paper with the official results.  Bi-Sent2Vec has an F1 score of ~0.90 for fr-en and de-en language pairs, second to LASER by only a few percent, despite being a much simpler model. Our scores are also significantly better than the paraphrastic similarity paper you referred to. For ru-en, we obtain an F-1 score of 0.80, but the drop in performance is mostly due to the quality of the en-ru dataset used to train our embeddings.
>
>
> We agree that the results on MLDoc evaluation are quite mixed. However, our objective was to illustrate that Bi-Sent2Vec despite being a simple and computationally cheap method, it is on par with LASER when it comes to performance.
>
> We also agree that our claim that multilingual training can improve performance on monolingual tasks has already been shown before and we have added the relevant references to the revised version of the paper(see Page 9, 1st paragraph). However, amongst the joint-training methods, Bi-Sent2Vec also outperforms Transgrams and BiVec trained on the same corpora by a significant margin, again hinting at the superiority of the sentence level loss function over a fixed context window loss for monolingual quality.
>
> While Kementchedjhieva et al. (EMNLP 2019) discuss the problem with the MUSE datasets and the saturation of performance on them as the training dataset size increases, Figure 2 also shows the saturation of performance on the sentence retrieval tasks along with the MUSE word similarity datasets which does point towards the argument made by us.
>
> While the extension from Sent2Vec is simple, we respectfully disagree that it is an incremental work. All the previous jointly trained methods for word embeddings showed improvement on cross-lingual word translation quality. For language pairs which are close, our models outperform all these jointly trained methods in two spheres (cross lingual sentence retrieval and monolingual word quality) by a significant margin and show similar performance on the cross-lingual word translation. This indicates that our sentence-level training objective promulgates better compositionality characteristics in our word embeddings as compared to earlier methods.  For dis-similar languages, the improvements are even more pronounced and results on cross-lingual word translations are also state-of-the-art showing the improvement made by our method on competing jointly trained cross-lingual word embedding methods. We thank you for your recommendations which have helped improve the quality of our paper.

---

### Official Review · AnonReviewer4 · 2019-10-29
**Official Blind Review #4**

**Rating:** 8

**Review:**

* Recommendation
While the contributions in this work are not staggeringly innovative, they are well grounded in existing work and well supported by experiments. Therefore I think the paper should be accepted.

* Summarize paper’s major contributions

The authors aimed to improve on the task of cross-lingual sentence retrieval, by introducing a model with a modified objective function, which utilizes a cross-lingual loss. They demonstrated that this objective function led to large improvements on word-level representation tasks and cross-lingual sentence retrieval, and achieves competitive performance on a document-level task while being more computationally efficient. The authors performed an in-depth ablation study, to support their claims that the proposed model addresses some of the key problems with other existing approaches to cross-lingual representation learning (e.g. hubness).

* Comments on the paper

This paper is exceptionally well written, organized, and clear. In addition to a solid introduction and related works sections, which frame the problem nicely, the conducted experiments thoroughly demonstrate the performance of the proposed model, as they evaluate on several granularities (word, sentence, document), as well as a robust ablation study and analysis. By the end of the paper, I am sufficiently convinced by the work and its contributions.

Meanwhile, I believe the paper could benefit from more discussion or analysis of cases where the proposed model did not lead to improvements, in particular with Russian in the word-translation retrieval experiment, where TRANSGRAM outperforms the proposed model. Although the authors briefly note this in the Discussion section, there is unfortunately no conversation about why this may be. The proposed model becomes less convincing when I consider that it might only work for other agglutinative, English-like languages, and I wonder how this approach would fair with other morphologically rich languages similar to Russian, and non-agglutinative languages in general.

* Minor corrections:

- In Figure-1’s caption, at the very end, there is a space missing between “sentence_(cross-lingual compotent).”

- Some missing colon’s (:) throughout the paper when breaking from a paragraph to introduce a list (e.g. “Contributions” in the Introductions)

- The x-axis of Figure 2 and Figure 3 are unclear to me, and a bit difficult to read. How do I interpret “10^-2” as a corpus size? In other words, what are “10^-2 amounts of data.” Fix this.

Overall, great work. I also appreciate the details about training both in the paper and appendix, that will be useful for those wishing to reproduce this work.

* Questions for authors

- Why do you think that TRANSGRAM outperformed your system on the word-translation retrieval experiment for Russian? Do you have any reasons to believe that the proposed model cannot extend well to other morphologically rich languages, or languages very dissimilar to English?

**Experience Assessment:**

I have read many papers in this area.

**Review Assessment: Checking Correctness Of Derivations And Theory:**

N/A

**Review Assessment: Checking Correctness Of Experiments:**

I carefully checked the experiments.

**Review Assessment: Thoroughness In Paper Reading:**

I read the paper thoroughly.

---

> ### Author Response · Authors · 2019-11-15
> **Answer to Official Blind Review #4**
>
> We thank the reviewer for the feedback and comments.
> We will incorporate the formatting suggestions as suggested by the reviewer.
>
> > Regarding the performance of Bi-Sent2Vec on the English - Russian corpus and languages dis-similar to English.
>
> We used a different dataset of a different nature for the English - Russian pair. It was mostly composed of parallel subtitles which in many cases are only loosely aligned and not exact translations of each other as compared to ParaCrawl used for other language pairs.
>
> In order to ease your concerns further, we trained and tested Bi-Sent2Vec, TransGram as well as Bivec on the English - Finnish (different language families) ParaCrawl corpus v4 and obtained superior results to Bivec as well as TransGram on both cross-lingual word and sentence retrieval. Moreover,  the extent of improvement for these language pairs was larger than that for other language pairs signifying the usefulness of Bi-Sent2Vec on dis-similar language pairs . We have added these results to the revised version of the paper (Section 4.4 - Table 4) .

---

### Public Comment · ~Mozhi_Zhang1 · 2019-11-07
**Stronger BLI results for Procrustes+Refine**

Dear authors,

I want to point out that the Procrustes+Refine can be improved with simple normalization of monolingual embeddings: https://arxiv.org/pdf/1906.01622.pdf

For example, EN-RU can be improved to 52.1 after iterative normalization (in appendix of the paper). This doesn't seem to change the ranking of the methods, but it would make a stronger baseline.

---

> ### Author Response · Authors · 2019-11-13
> **Thank you**
>
> Hello,
> We thank you for your input. We agree with your suggestion and we'll incorporate the improved results in the paper.

---

### Decision · Program_Chairs · 2019-12-19

**Decision:**

Reject

**Comment:**

The authors propose a new approach to learning cross-lingual embeddings from parallel data. For an overview of this literature, see [0]. Reviews are mixed, and some objections seem unresolved. The authors also ignore a new line of research in which pretrained language models are used to align vocabularies across languages, e.g., [1-2] The paper would also benefit from a discussion of massively parallel resources such as JW300 and WikiMatrix. Finally, it feels odd not to compare to distilled representations from NMT architectures, e.g., [3].

[0] http://www.morganclaypoolpublishers.com/catalog_Orig/product_info.php?products_id=1419
[1] https://www.aclweb.org/anthology/N19-1162.pdf
[2] https://www.aclweb.org/anthology/K19-1004.pdf
[3] https://arxiv.org/abs/1901.07291